# Smart Multi-Sensor System for Remote Air Quality Monitoring Using Unmanned Aerial Vehicle and LoRaWAN

**DOI:** 10.3390/s22051706

**Published:** 2022-02-22

**Authors:** Rosa Camarillo-Escobedo, Jorge L. Flores, Pedro Marin-Montoya, Guillermo García-Torales, Juana M. Camarillo-Escobedo

**Affiliations:** 1Mechanic and Mechatronics Department, National Technological Institute La Laguna, Blvd. Revolución & Calz. Cuauhtemoc S/N, Torreon 27000, Coahuila, Mexico; rmcamarilloe@lalaguna.tecnm.mx (R.C.-E.); pcz97mm@gmail.com (P.M.-M.); 2Translational Biomedical Engineering Department, University of Guadalajara, Av. Revolución #1500, Guadalajara 44430, Jalisco, Mexico; garcia.torales@cucei.udg.mx; 3Electric and Electronics Department, National Technological Institute La Laguna, Blvd. Revolución & Calz. Cuauhtemoc S/N, Torreon 27000, Coahuila, Mexico; jmcamarilloe@correo.itlalaguna.edu.mx

**Keywords:** air quality, remote sensing, UAV, LoRa, smart sensors

## Abstract

Deaths caused by respiratory and cardiovascular diseases have increased by 10%. Every year, exposure to high levels of air pollution is the cause of 7 million premature deaths and the loss of healthy years of life. Air pollution is generally caused by the presence of CO, NO_2_, NH_3_, SO_2_, particulate matter PM_10_ and PM_2.5_, mainly emitted by economic activities in large metropolitan areas. The problem increases considerably in the absence of national regulations and the design, installation, and maintenance of an expensive air quality monitoring network. A smart multi-sensor system to monitor air quality is proposed in this work. The system uses an unmanned aerial vehicle and LoRa communication as an alternative for remote and in-situ atmospheric measurements. The instrumentation was integrated modularly as a node sensor to measure the concentration of carbon monoxide (CO), nitrogen dioxide (NO_2_), ammonia (NH_3_), sulfur dioxide (SO_2_), and suspended particulate mass PM_10_ and PM_2.5_. The optimal design of the multi-sensor system has been developed under the following constraints: A low weight, compact design, and low power consumption. The integration of the multi-sensor device, UAV, and LoRa communications as a single system adds aeeded flexibility to currently fixed monitoring stations.

## 1. Introduction

Deaths caused by respiratory and cardiovascular diseases have increased by 10%. Every year, exposure to air pollution is estimated to cause 7 million premature deaths resulting from problems ranging from respiratory infections to lung cancer, stroke and chronic obstructive pulmonary disorder [1]. In 2018, the World Health Organization (WHO) stated that 93% of the world’s children breathe polluted air every day. According to this information, 1.8 billion children breathe air so polluted that their health and development are in serious danger. In 2019, more than 90% of the global population lived in areas where pollution concentrations exceeded the 2005 WHO air quality guideline for long-term exposure to PM_2.5_ [2]. This increase in atmospherically active gases leads to soil and water acidification and the reduction of the vision of the atmosphere [3,4]. Since 2020, air pollution has been related to population mortality [5] and the rapid spread of COVID-19, focusing on cities whose PM daily concentrations were higher during the months preceding the pandemic than the annual average [6,7]. 

Mexico has specific areas with serious environmental problems. Torreón city, located in Region Lagunera of Coahuila state, was ranked as the fourth highest of 248 of the most contaminated regions around the world in terms of ammonia concentration [8]. Ammonia is not usually considered in the different official standards due to its small presence in natural conditions, being observed in high quantities in industrial regions where substances or compounds related to this material are improperly handled and during intensive agricultural and livestock activity, which promotes the release of greater than average amounts of fertilizer use. Additionally, when combined with other acidic contaminants such as sulfur dioxide (SO_2_) and nitrogen oxides (NOx), ammonia (NH_3_) becomes ammonium (NH_4_) +, which has a high diffusion range, extending several hundred kilometers from its point of origin (between 100 and 1000 km) [9,10].

Previously, the measurement of air quality was carried out by non-continuous methods known as “wet chemistry” for gases and high volumes for suspended particles using complex and costly stationary equipment to “collect” data [11]. This paradigm is changing, with advanced electronics and new low-cost and easy-to-use sensors being developed, providing high-resolution and real-time data that can be “accessed” [12,13]. These attributes offer opportunities to improve a range of existing air pollution monitoring capabilities and provide avenues for new applications. The integration of emerging technologies, such as unmanned aerial vehicles (UAV), wireless sensors networks (WSN), the internet of things (IoT), and advances in computing and communications (TIC’s), achieves availability and accessibility of data in real-time, as well as expanding the sampling areas [14,15] and generating instant alerts when the peak values are exceeded [16]. 

Several solutions have been proposed for air quality monitoring and measuring environmental parameters in this sense. Some developed systems exploit wireless communication for the development of WSN. For example, Kasar et al. [17] presented a low-cost WSN-based Air Pollution System using the ZigBee protocol and an Arduino to monitor CO_2_ and NO_2_. However, the range for these modules does not exceed one hundred meters, and the battery consumption is high. In contrast, Lee and Ke [16] and Yousuf et al. [18] implemented a LoRa WAN network to collect data from IoT sensors for a large geographical area. The results proved the communications between nodes, but these did not acquire data from the sensors. An improved proposal was made by Candia et al. [19], who present a monitoring system of air quality based on the LoRaWAN network with low-cost sensors to measure PM_10_ and PM_2.5_, and Raju. et al. [20] provides an overview of LoRa in analyzing the air pollutants data from different sensors (CO_2_, NO, CO, temperature, and humidity) connected to a PC to store and analyze the real-time data for further use. Liu et al. [21] researched monitoring with three low-cost sensors for PM_2.5_ particulate matter, obtaining results very close to those detected by the environmental monitoring system. Samad et al. [22] highlight that low-cost sensors can be reliable for air monitoring systems, as long as calibration tests are carried out, considering the influence of environmental variables such as temperature and humidity. Badura et al. [23] evaluated three small, low-cost optical sensors that could be used to improve the spatial and temporal resolution of PM data.

Drones, or unmanned aerial vehicles (UAVs), have become technologies employed to collect data in a given outdoor environment. In this sense, Zulkifli et al. presented a simple monitoring system based on the Internet of Things (IoT). A MQ 135 sensor was mounted on a drone to transmit dates to a smartphone, using the Blynk application, air quality level was measured in ppm [24]. Wivou et al. proposed a system to collect data through sensors mounted in a drone. The control system was based on an Arduino to acquire three MQ-type sensors to monitor CH_4_, CO, and CO_2_ [25]. Trasviña-Moreno et al. [26] proposed a WSN using a UAV as a mobile data collector for monitoring marine environments. The sensor node was based on temperature, humidity and pressure sensors, anemometers, and an insulated temperature sensor for water measurements. The control was based on the Beagle Bone Blac (BBB) board AM335x and the WSN was implemented with the LoRa NET. Madokoro et al. [27] developed a unified sensor and communication system for in situ atmospheric measurements. The sensor system is mounted on a drone to measure PM_2.5_ and LoRaWAN was used for the wireless communications. A single-board computer SBC Raspberry for the control and a mobile battery was used. 

Although the previous developments have presented significant advances in the monitoring of air pollutants, it is difficult to generate a broad overview of environmental conditions and health risks, if not fully involve the main air pollutants or ecological parameters in the same sampling or monitoring system. Nowadays, drones have been used to offer greater mobility to the measurement systems, these being limited by their payload. As described above, the selection of development platforms for control (Arduino, Raspberry, BBB, etc.), and some sensors increased the weight and dimensions of the system, e.g., more than 1 kg. Expensive high-end UAVs offer this payload capacity. In addition, the validation of a system, its characteristics and accuracy were not compared with fixed stations or international systems to have reliability in the measured data. 

In the present work, we presented the development and implementation of a smart multi-sensor system for remote monitoring air quality. The proposed system consists of a sensor node based on microdevices with the capacity to measure in-situ the concentrations of carbon monoxide (CO), nitrogen dioxide (NO_2_), sulfur dioxide (SO_2_), ammonia (NH_3_), and suspended particles PM_10_ and PM_2.5_, as well as environmental parameters such temperature, humidity, and pressure. Due to its miniaturization and low cost, the system could be operated as a fixed or mobile station for remote sensing applications using LoRa WAN protocol and an Unmanned Aerial Vehicle (UAV), which collects data in real-time. The device was probed by measurements carried out in three principal metropolitan areas and agriculture and livestock zones in Mexico. The results were compared with the official data and international databases.

## 2. International Standards and Monitoring Networks by Environmental Protection

In order to measure air quality and environmental conditions, international standards, measurement methodology [11] and prevention parameters have been implemented to reduce the risk of the population and improve the environment. 

The National Ambient Air Quality Standards (NAAQS) were established to protect public health in the United States. This standard specifies the concentration levels allowed, over a set period of time, for carbon monoxide (CO), lead (Pb), nitrogen dioxide (NO_2_), sulfur dioxide (SO_2_), ozone (O_3_), particulate lead, and suspended particulate mass (PM_10_, PM_2.5_) [11]. 

The Mexican Standard Metropolitan Air Quality Index (IMECA, for its acronym in Spanish) was created in 1982 and it was initially based on the NAAQS. Due to the growing problem of air pollution in Mexico City, the IMECA decided to relax the limits of health risk and now it has more permissible threshold values than NAAQS because according to NAAQS, Mexico City would always be in an environmental contingency. One aspect to consider is that IMECA indicated only the value of the pollutant that presents the greatest risk at this moment, with it being possible that other pollutants’ concentrations can also be found at dangerous levels without being reflected in the final value [11]. 

The Air Quality Index (AQI) is also used to indicate how pure the air is or how much pollution is in the air and the health effects on the population. 

Each country provides it is own AQI. The AQI is managed by the Environmental Protection Agency (EPA) in the USA and looks for five major air pollutants: particle pollution, ground-level ozone, carbon monoxide, nitrogen dioxide, and sulfur dioxide. Table 1 shows the numeric values from the AQI.

To obtain the AQI values, the EPA collects data from monitoring stations and calculates them according to [29] to provide simple information about the local air quality.

With increasing public demands for timely and accurate air pollution reporting, more air quality monitoring stations have been deployed by governments and society in urban metropolises to increase the coverage of urban air pollution monitoring.

In this sense, international air quality monitoring systems have been developed to inform the population about air quality. Plumelabs integrates a portable system called Flow, which uses a particle counter to measure the concentrations of particulate material through light scattering and a metal oxide sensor to measure the concentrations of gases. The equipment measures particulate matter (PM_1_, PM_2.5_, and PM_10_), volatile organic compounds and nitrogen dioxide. The information is displayed on an interactive map called Plume Air Report and a mobile application with indicators representing the AQI scale. Another platform for measuring the air quality monitoring is The Weather Channel which, in an alliance with the IBM Cloud Service and the use of artificial intelligence, data, and cloud computing [30], forecasts air quality. Its web page is weather.com and it displays the environmental conditions, air quality, and the AQI scale. National Air Quality Information System (SINAICA, for its acronym in Spanish) collects, transmits, and publishes information about air quality in Mexico. It is generated on the monitoring systems of air quality (SMCA for its acronym in Spanish) spread all over the country. A total of 36 SMCA with 242 monitoring/samples systems across the country have been operating according to NOM156 [31]. SINAICA presents a problem due to the constant interruptions in the service, which generates unreliable data in its reports.

Between the pollution measurements at the regional level and the national measurements, the uncertainty in the estimation of emissions is too high, leading to the conclusion that measurement campaigns should be carried out in the field [32]. In addition, although monitoring stations are installed in Mexico, the absence of national regulations for the design and installation of air quality monitoring networks means that air quality monitoring networks have different operational criteria and have strong limitations in terms of the availability of financial, human, and material resources. For this reason, the design of new networks to measure air quality, as well as the redesign of current networks and the expansion of their coverage, are necessary to improve their operation, by homogenizing the criteria of their establishment, process, and maintenance [33] as well as mobility.

## 3. Proposed System

### 3.1. System Architecture

The smart multi-sensor system architecture is shown in Figure 1, to determine the influence of pollutant emissions on air quality. The system comprises three elements: a smart multi-sensor (sensor node), the LoRa wireless communication module (gateway) and the visualization and processing data (base station). The sensor node is mounted on a UAV for collecting data across a large geographical area as a remote fixed station or mobile station. 

The multi-sensor node integrates a microcontroller and a LoRa communication module RA01. It is configured as an end-device. The gateway uses the same LoRa communication module RA01. The end devices send the information to the gateways in a process known as uplinks. These send the information to the network servers and thus to its endpoint, which is an application itself. Similarly, network servers can send messages through gateways to end nodes in a process called downlinking. This generates two-way communication. The method of uplinking is shown in Figure 2.

To better explain the full functionality of this proposed device, this section has been divided into the following subsections, covering the features of the multi-sensor node, the LoRa wireless communications and the description of the UAV, including the design and manufacturing of the prototype support for mounting the sensor on the drone. Moreover, the methodology and some gas considerations are also covered.

### 3.2. Smart Multi-Sensor Node

The smart multi-sensor node was developed to measure and monitor the most important pollutant gases concentration. The proposed system was integrated modularly to develop the four following measures: (a) meteorological parameters, (b) pollutants gases, (c) SO_2_, and (d) particulate material. 

A BME280 sensor (Bosh Sensortec GmbH, Reutlingen, Germany) was integrated to measure temperature, relative humidity, altitude, and atmospheric pressure, with a resolution of 0.01. It is the importance of measure these variables because they affect the measurement and are a reference for the calibration of the sensor device [22]. 

In this same module, Grove-Multichannel Gas Sensor MICS-6814 (SGX Sensortech, Nürnberg, Germany) is a robust MEMS sensor with the ability to detect carbon monoxide (CO) range of 1–1000 ppm with a sensitivity factor of 1.5, nitrogen dioxide (NO_2_) range of 0.05–10 ppm with a sensitivity factor of 2, ammonia (NH_3_) range of 1–500 ppm with a sensitivity factor of 1.5. For polluting gas monitoring the sensor has a sensitivity factor between 1.2 and 2. Other gases such as hydrogen (H_2_), methane (CH_4_), propane, ethanol and iso-butane could also be measured. The gas sensor was configured according to [34]. Its structure consists of an accurately micro machin ed diaphragm with an embedded heating resistor and the sensing layer on top. The gas sensor includes three sensor chips with independent heaters and sensitive layers. One sensor chip detects oxidizing gases (OX), the other sensor detects reducing gases (RED), and the other detects NH_3_. 

In addition, the ULPSM-SO_2_ sensor (SPEC Sensors, Newark, CA, USA) was integrated to detect sulfur dioxide (SO_2_) in a range of 0–20 ppm with a resolution of 1.5 μm and reading accuracy of ±2%. 

After the monitoring of pollutants, the multi-sensor system measures particulate matter, integrating a digital particle sensor module based on the PMS7003 device (Plantower, CHN) for the measurement of 10 and 2.5 micron particles sizes, in a range of 0–500 μg/m^3^, signal response of less than 1 s with resolution of 1–2.5 μm and 2.5–10 μm. The particle counting efficiency is 98%. 

For air quality measurement according to the methodology proposed here, the acquisition of signals begins by calibrating the sensors by “heating” for 30 min the first time of operation and 5 min for subsequent operations.

Five parameters are important for a gas sensor based on the change of its resistance: response, response time, recovery time, selectivity, and working temperature. The response to reductive gas is defined as [35]:(1)$$S=R_{a}/R_{g}$$
or [36]:(2)$$S=\left(R_{a}-R_{g}\right)/R_{g}$$
where *Ra* and *Rg* are the resistances of the device in air and the target gas atmosphere, respectively. Generally, the response time (Tres) and recovery time (Trec) is defined as time spent by a sensor to achieve 90% of the total resistance change during the adsorption and desorption process respectively. An optimal gas sensor should meet the requirements of large sensing response, low working temperature, and high selectivity to the target analytes [37]. The characterization of the multi-sensor system for each gas was carried out in preliminary work [38].

To control the microdevices and data processing the Teensy 4.0 development board (SparkFun Electronics, Niwot, CO, USA) was used, which integrates an ARM Cortex-M7 processor at 600 MHz, with an NXP iMXRT1062 chip and 12 A/D converter. Figure 1 emphasizes the exchange of data between the control and the sensors as well as the multiple operation and communication interfaces (Analogic, I^2^C, Serial, SDIO, SPI, and WIFI). The control system restricts the communication distance to a maximum of 122 m of height and a maximum of 457 m in the horizontal direction, according to NOM-107-SCT3-2019. The multi-sensor node was configured as an end-device integrating a LoRa communication module RA01 (Semtech Co., Camarillo, CA, USA).

For the supply and operation of the system, a 7.4 V LiPo battery, 300 mA hr (Turnigy, Kwun Tong), adjustable voltage regulator, 3.3 V regulator, and a pair of indicator LEDs was integrated into the smart multi-sensor node.

The multi-sensor system implemented is shown in Figure 3. The distribution of components allows an easy operation of the system and maximum compaction to not interfere with the development of the flight. For its implementation, low-consumption, long-range, surface-mount devices were selected and modules for its connectivity, obtaining a total weight of 82 g. Due to its modular design, components can be removed and easily replaced in failure.

### 3.3. LoRa Wireless Communication

LoRa is an RF modulation technology for low-power, wide-area networks (LPWANs). This technology enables the extremely long-range data links’ communications: up to 5 km in urban areas, and up to 15 km or more in rural areas (line of sight). The LoRa RA01-SX1278 operates on the ISM band of 433 MHz, has an idle current consumption of 1.6 mA and a current working consumption of 4.5 mA with an operating voltage of 3.3 V [39,40,41]. LoRa can be used in applications that require long-range or deep in-building communication among many devices with low power requirements and that collect small amounts of data. LoRa is typically used in IoT. An ESP32 board (Esspresif Systems Co., Shanghai, China) was used for the communication control. It integrates a Tensilica Xtensa LX6 dual-core microprocessor, and a resolution of 12 bits as an SoC technology. It is used for mobile applications and the internet of things. The ESP32 provides connectivity to link to a network service (WiFi, Bluetooth, and BLE (Bluetooth Low Energy)) and SPI protocol. 

### 3.4. System Transportation

#### 3.4.1. Unmanned Aerial Vehicle

A Mavic Air 2 unmanned aerial vehicle (UAV) was used to provide mobility to the proposed system. The UAV is equipped with a three-axis stabilization system, a global navigation satellite system with GPS + GLONASS, a maximum transmission distance of 10 km on the 2.4 and 5 GHz band, 34 min of maximum flight time, and a weight of 570 g. The system includes three modes of flight control: Normal, Sport, and Tripod. The Advanced Pilot Assistance Systems 3.0 (APAS 3.0) is enabled when using the Normal Mode and helps avoid obstacles easier. It includes a smart return to home (RTH) function which brings the aircraft back to a previously established Home Point. The UAV is equipped with an Infrared Sensing System, which consists of two 3D infrared modules and a Forward, Backward, and Downward Vision System that consists of two cameras and provides the UAV with a detection range of 0.35–22 m for the Forward Vision System, 0.37–23.6 m for the Backward Vision System and an altitude of 0.5 to 30 for the Downward Vision System. Although these UAV are not designed to carry weight, the Mavic Air 2 can an additional 830 g of payload. This feature gives us greater freedom when designing the multi-sensor system in terms of weight.

#### 3.4.2. Support Structure

The multi-sensor system was modularly integrated, and it was necessary to manufacture a support structure to mount it on the drone. This structure was designed using SOLIDWORKS^®^ software. The design is shown in Figure 4, and has two parts: a support base to keep the electronic instrumentation and a supported lock with the function of closing the base door support to protect the sensor node. This structure was manufactured using the 3D printer Prusa i3 mk2.5s (Prusa3d, CR) using ABS filament, which is more resistant than the commonly used PLA filament. 

The multi-sensor system placed on the support structure was mounted on the drone as shown in Figure 5. The final dimensions were 41 × 80 × 49 mm, adding a total weight of 0.92 g to the drone so it did not generate instability in the flight.

### 3.5. Methodology

Before operating the sensing device, it must be ensured that the backup storage unit is installed in the transmitting node and the battery is 100% charged in the device and UAV. The first time the device is operated, a calibration of the gas sensors should be performed, “heating” for 30 min in relatively clean ambient conditions. After this start step, each time the sensor is used, pre-heating must be carried out on the gas sensors, “heating” for 5 min before taking the measurement. Once this time has elapsed, the sensor node (end device) is ready to start the measurement by pressing the power button. At this time, the Gateway device connects to a network, opening a channel for data reception. During this waiting time, the device is placed on the drone Mavic Air 2 (DJI, Pekin, China), using the adapter support structure and the scanning of measurements at different heights begins. An algorithm for data acquisition, processing, storage, and transmission was implemented, which generates a 14-bit vector, according to the order of four following measures: (a) meteorological conditions, (b) pollutants gases, (c) SO_2_, and (d) particulate material. All data obtained are stored in the micro SD memory and sent to the Gateway in data packets separated by an identifier character. The data received are displayed in an API provided by Matlab (MathWorks Inc., Natick, MA, USA) called ThingSpeak for visualization in real time. They are exported in .CSV files and can be linked to an email to communicate values oriented to applications in IoT. This process lasts approximately 6 s, which is repeated until the system is turned off. The data were obtained from the API of ThingSpeak.

### 3.6. Gas Considerations

In order to measure these air pollutants, one must know the emission sources as well as their behavior.

SO_2_ pollution was worsened by increased emissions caused by rapid urbanization and industrialization. The dispersion of SO_2_ concentrations was influenced by the meteorological parameters of wind speed and direction, temperature, and relative humidity. Multiple regression models showed that SO_2_ concentrations increased with the decrease of wind speed and temperature, and with the increase of relative humidity [42]. NO_X_ is a group of gases formed by nitrogen and oxygen (NO_2_, NO, etc.) emitted from the burning of fossil fuels in road vehicles and power generation plants. NO_X_ participates in ozone formation through photochemical reactions, and their half-life varies from a few hours on sunny days to several days in humid periods. Both NO_2_ and SO_2_ react with water in the atmosphere to form nitrates and sulfates, components of acid rain [43]. The increase of gaseous ammonia (NH_3_) concentration in the atmosphere significantly impacts the regional air quality, human health, and the nitrogen cycle of ecosystems. This shows a significant increasing trend at a rate of densely populated, intensive agricultural activities. NH_3_ concentrations show their highest values in summer and lowest in autumn. Such seasonal variation is mainly affected by seasonal differences in NH_3_ emissions and meteorological conditions. Control measures show that they can reduce SO_2_ and NO_2_ pollution but have not yet mitigated atmospheric NH_3_ pollution [44]. There is also little effect on the PM_2.5_ concentration due to ammonia emissions variability in the summer when gas-phase changes are favored, but variability in wintertime emissions, as well as in early spring and late fall, will have a larger impact on PM_2.5_ formation [45].

On the other hand, the variations of mass concentrations of PM_2.5_, PM_10_, SO_2_, NO_2_, CO, and O_3_ were analyzed based on data from 286 monitoring sites obtained for one year. By comparing the pollutant concentrations over this length of time, the characteristics of variations of the mass concentrations of air pollutants were determined using the Pearson correlation coefficient, which establishes that a relationship between PM_2.5_, PM_10_, and the gas pollutants exists [46,47,48]. A lot of studies have provided transferable models to estimate the health effects of air pollutants to support the creation of environmental health public policies for national and international intervention [49]. 

In this sense, environmental variables such as temperature, humidity, wind speed, direction, and the weather and seasons affect the concentration of air pollutants.

## 4. Location Measurement Site Selection

A very important factor in establishing effective air quality monitoring programs is to assign the optimal location for monitoring stations based on five criteria (population, wind direction, spatial proximity to roads, industries, and high traffic areas) that are considered the most important criteria [50]. Three important metropolitan regions of Mexico were selected to monitor according to their pollution indexes: Region Lagunera Coahuila-Durango, metropolitan area of Monterrey, N.L and the metropolitan area of Guadalajara, Jal. The monitoring sites are shown in Figure 6.

Torreon is a Mexican city belonging to the state of Coahuila, and is the 10th largest city in Mexico with an average altitude of 1120 m.a.s.l. Torreon is subject to occasional harsh climatic conditions such as low temperatures, strong winds and drought in a season-dependent manner. Annual average temperatures are typically 2 °C to 5 °C for the winter months of December and January, and 39 °C to 45 °C for June and August. The city has a population of 731,902 residents [51]. Its main source of contamination is the PEÑOLES metallurgical industry, the FERTIREY fertilizer industry, and the LALA dairy industry, including livestock and forage agriculture. Torreon currently has only one air quality monitoring station CONALEP, and is not in operation [52]. The location of the monitoring site 25°32′07″ N 103°26′06″ W.

The Matamoros city in Region Lagunera of Coahuila has a population of 119,919 residents [51]. The city has an average altitude of 1120 m.a.s.l. Its climate is hot, with rains in summer and strong winds that reach 44 km per hour in spring produce dustbins. The annual average temperature oscillates between 22 and 24 degrees Celsius in summer, with records of up to 40 °C to 53 °C. Winters have been recorded with minimum temperatures of −3 to −8 °C. The sources of contamination come from the brick industry, burning mesquite charcoal, and livestock for milk production and forage agriculture [53]. The location of the monitoring site is 103°13′42″ W 25°31′41″ N.

The Gómez Palacio city in the Region Lagunera of Durango has a population of 371,002 residents. Its climate is desert. The average annual temperature is 22.6°. In a year, the rainfall is 225 mm. Gomez Palacio, currently has two fixed monitoring stations; El Campestre and La Esperanza [52], with intermittent operation. The main sources of contamination are found in the manufacturing industry, thermoelectric and combined cycle plants, extraction of construction materials, marble industry, and the livestock sector. The location of the monitoring site is 25°34′40″ N 103°29′54″ W. 

The other metropolitan area is because Monterrey, Nuevo León, currently considered the second largest in Mexico. The city has a population of 5,341,171 residents [54]. The climate averages 23 °C annually, with light wind gusts. The temperature contrast throughout the day can be very noticeable and this is due to the fact that the region is 500 m.a.s.l. This metropolitan area has 14 fixed monitoring stations [52] that constantly warn about a large amount of suspended particles less than 10 µm present in the region [55]. The primary sources of emissions are the oil sector, petrochemicals, the glass industry, cargo transport (heavy and light), and automobiles [56]. The location of the monitoring site is 25°40′00″ N 100°18′00″ W.

The metropolitan area of Guadalajara, Jalisco, has a population of 5,268,642 residents [57]. The region is located above 1500 m.a.s.l. and its climate is mainly humid. This metropolitan area has ten fixed monitoring stations [52]. The air pollutants constantly reported are suspended particles less than 10 µm and Ozone (O3) [58]. Its main source of pollution comes from agricultural and garbage burning, industrial emissions (food, cement and chemical), livestock farms, vehicular traffic (where almost three vehicles per inhabitant are reported), and brickyards [59]. The location of the monitoring site is 25°40′00″ N 100°18′00″ W.

## 5. Experimental Results

### 5.1. Experimental Set Up

The multi-sensor system was probed by tests carried out in different sites of Mexico. The monitoring time was determined according to a set of heights, which were previously programmed in the flight plan of the drone, and its battery time for a safe flight of approximately 25 min. Six navigation heights were set: home, 10 m, 30 m, 50 m, and 100 m. This protocol was conditioned by weather conditions and flight restrictions in the measurement areas. For the data acquisition, the setup of the system is carried out according to the proposed methodology. The multi-sensor system was put on the drone and then it flew vertically during a range of 16 to 26 min. Two frequencies were programed, every 6 and 1.65 s. After the measure, the results were compared in real time with international information systems, Weather.com [60] and Plumelabs.com [61]. ThenNational air quality information system (SINAICA) from the government website [52] did not offer data because it was frequently out of service. Meteorological parameters’ data were validated by the National Meteorology System (SMN by its acronym in Spanish), and were downloaded from the government website [62]. The multi-sensor system mounted and flying over UAV is show in Figure 7.

### 5.2. Monitoring in Region Lagunera Coahuila-Durango

Five measurement sites were selected in the Region Lagunera and are shown in Figure 8, considering the proximity to the pollution sources.

#### 5.2.1. Torreon, Coah

La Laguna Technological Institute monitoring site was selected. This site is located very close to the most important metallurgical industries in Latin America and fourth worldwide, Met-Mex-Peñoles and Fertirrey which manufactures fertilizers rich in nitrogen and sulfur and it is the most important ammonium sulfate production plant in the north of Mexico. The tests were carried out last winter during January, the season with the highest level of pollution [38]. 

In this series of experiments, according to the algorithm for data acquisition, meteorological parameters for two days were measured, and these are shown in Table 2. The acquired data is very close to those acquired by Gov. Website, with a difference in relative humidity measurements of 3 to 5%, and of the temperature of 1 to 4 °C. It can be seen that the measurements by the two systems show a relationship of the low humidity-high temperature, and on the contrary, low temperature-high humidity, which is a climate characteristic of this region. 

For air quality, we monitored the concentration of SO_2_, NH_3_, CO, NO_2_, PM_10_, and PM_2.5_ in the site location. Every 6 s, 168 to 250 samples were acquired average in simple data. The response of the multi-sensor system is shown in Figure 9 with data acquired on 11 January 2021, and Figure 10 shows the values measuring on 15 January. 

In Figure 9a and Figure 10a one can be seen that the NH_3_ sensor shows a delay. The sensor needs a period of time to set up its chemical equilibrium. This is due to chemical compounds being desorbed or absorbed on the sensing surface after which point the resistance will stabilize. Generally speaking, the longer the warm-up phase, the better the precision will be. After that time the measurement stabilizes. In the case of the CO sensor, it did not present sensitivity which is consistent with what was reported in [34]. On the other hand, it can also be seen that the suspended particles, in this case, study, remain suspended up to the height of the drone.

After the analysis, the results were compared with data collected from international air quality measurement systems. This comparison is shown in Table 3. According to the precision and linearly of an optical sensor of PM, coefficients of variation were below 7% [23].

In these experiments, the values provided by the proposed system correspond to real-time measurement at specific points in situ. On the other hand, the data reported by the database corresponds to a 24 h average, considering a regional area through satellites and atmospheric behavior algorithms. Figure 11 shows capture images of the drone where it can be seen that Figure 11a presence of dust at the monitoring site and Figure 11b the contamination by industry. In this case, we can prove that the results measured were real-time and only represent the 1.7% (25 min) of the total period measured (24 h measured by the international air quality systems.

#### 5.2.2. Matamoros, Coahuila

In the second series of experiments in the Region Lagunera, data acquisition was measured in the agricultural and livestock areas. In order to validate the multi-sensor monitoring system, controlled tests were made to measure CO, NO_2_, NH_3_, and PM_2.5_. For collecting data, there were restrictions by the farmers’ property, mainly that taking pictures in flight or on the ground and revealing the location was not allowed. Monitoring was carried out at a distance of 1 to 3 m from the crop fields, in the last autumn-winter cycle, during February. In Figure 12 we can see the results in the measurement of NH_3_ and NO_2_ and one can observe that there are no health risks, considering that both measurements are within the levels allowed by the WHO. Then, we were informed that nitrogen fertilizers had not been used.

The response of the proposed system at NO_2_, showed an oscillation between 0.05 and 0.06 due to the sensitivity of the sensor. Its measurements were low and there was no risk to health. In the case of the measurement of suspended particles (PM_2.5_), two significant results were observed; the readings that were taken on the second day were much higher because the climate conditions were dry and dusty. The sensor measurement showed an unhealthy quality range according to the AQI. This is a very constant condition in this city. 

The second series of measurements were carried out in small livestock for milk production with 25 cows. For this measurement, preventive actions were considered, due to the direct exposure of the cow’s waste, which emits polluting gases such as ammonia, mainly found in cow’s urine. N95 masks were used as protection. In Figure 13 one can observe that NH_3_ increases as a function of the height when the drone rises. For example, a height of 3 m, the NH_3_ concentration was approximately of 5 ppm. While at 15 m the NH_3_ concentration was approximately of 5 ppm. Both measurements are within the permitted level of 25 ppm, above which there are health risks a risk. 

The results showed the sensitivity of the proposed system to NH_3_. Although there was no access to larger barn facilities, it can be observed that cow waste (due to their feed) increased NH_3_ values. This is verified by comparing the results obtained in the measurement of NH_3_ of an agricultural plot with the values obtained in a small livestock.

Although the results were not compared with national or international systems because the NH_3_ pollutant was not measured by them, we can observe that the proposed system was sensitive to NH_3_ concentration. Furthermore, we found a higher concentration of ammonia in stables than the concentrations measured in critical zones of metropolitan areas. According to the observed results, there could be a correlation between the generation of NH_3_ and the cattle in stables, according to [8]. 

#### 5.2.3. Gomez Palacio, Durango

The third series of experiments in this region was carried out in Gomez Palacio, Durango. Monitoring was carried out for nine continuous hours for six days from 21 to 28 September, as a fixed monitoring system. The frequency sampling was 1.6 s. The site selected to monitor was situated on the periphery of the town near a stable field and walnut trees (nogalera).

Firstly, meteorological parameters were measured by the multi-sensor system, and the data are shown in Table 4. The acquired data are very close to those acquired by our referenced government website. 

Simultaneously, we measured the pollution conditions, and more than 20,000 samples per day were acquired as a result of this monitoring. 

Figure 14 shows the evolution of air pollution in this site according to the environmental parameters described in Table 4. The weather in 2021 was typical for early autumn with representative gusts of wind, as well as high temperatures and humidity.

In Figure 14, we can observe in Figure 14a that the proposed system was sensitive to the measurement of SO_2_ concentration. These concentration increases are located in the time from 15:00 to 18:00 h. This behavior was constant during the 6 days of study. The data acquired for NO_2_, shown in Figure 14b, displayed the same behavior, this presented an increasing trend with a higher concentration over time from 12:00 to 15 h. The results for NH_3_ measurement was shown in Figure 14c. The NH_3_ sensor, after auto-calibration, had its output values stabilized at around 0.91 ppm. 

In this sense, with a wide range of monitoring samples (20,000 samples per day), during a year, the proposed system could define a forecast in environmental behavior, defining an approximation of air quality. The output values were unsteady with wide fluctuations due to the sensibility of the sensor. In this case, moving averages as a statistical prediction technique was used to reduce the impact of irregular data in a time series of data. Figure 15 depicts the data acquired from the particulate matters sensor as an adjusted graph. On 25 September, the mean value from the international monitoring system during this period was 5.85 ± 1 µg/m^3^ to PM_2.5_. The mean value obtained using our system during this period was 6.5 µg/m^3^. A comparison of both values shows the difference as 0.65 µg/m^3^. 

In Figure 15, we can observe that particulate matter in the air was influenced by meteorological conditions. In particular, lower concentrations correspond to measurements when there were high gusts of wind, see Figure 15a,b. 

The results are shown in Table 5. SINAICA was unavailable due to technical difficulties with the website.

Figure 16 shows capture images of the drone where the presence of dust at the monitoring site can be seen. 

### 5.3. Monitoring in Monterrey Metropolitan Area

In the Monterrey metropolitan area, the weather conditions and pollution were measured during four days near to three different air quality monitoring stations, as shown in Figure 17. However, due to weather conditions and the denied permits to fly the drone, most of the tests were performed keeping the system static.

The meteorological parameters present during the measurement can be seen in Table 6. Slight variations can be observed in the sensor readings with respect to the information provided by the government; this is mainly due to the mountainous terrain of the region.

Figure 18 illustrates the acquired data by the multi-sensor system due to concentrations of SO_2_, NH_3_, CO, NO_2_, PM_10,_ and PM_2.5_ on 15 July 2021. Figure 19 shows the values measured on 16 July 2021. In this experiment, the multi-sensor system was evaluated by acquiring data at different heights and keeping the system at a fixed height. In the case of the system when it is on the drone, it can be seen that the values of SO_2_ and NH_3_ show a decrease as a function of the height, as it is shown in Figure 18a. In Figure 19a, the values of SO_2_ and NH_3_ show a constant trend at a fixed height.

Table 7 shows the comparison of the multi-sensor system with respect to an international source and the data obtained at the same time from the network of the corresponding SINAICA monitoring stations.

Figure 20 shows captured images of the drone where the presence of dust can be seen at the monitoring sites: Figure 20a for Obispado station and Figure 20b for San Nicolas.

### 5.4. Monitoring in Guadalajara Metropolitan Area

In the Guadalajara metropolitan area, the meteorological parameters and pollution were measured during four days near to three different air quality monitoring stations, as shown in Figure 21. However, due to weather conditions and the denied permits to fly the drone, most of the tests were performed keeping the system static. Although this metropolitan city has 10 monitoring stations, only a couple of them measure three or more pollutants, which represents a shortage of information.

The meteorological parameters present during the measurement can be seen in Table 8. The Centro and Vallarta stations only provide the temperature inside the monitoring station; they do not take into account the outdoor conditions over which the monitoring is performed.

Figure 22 displays acquired data near to in Centro station on 24 November 2021 for the concentrations of SO_2_, NH_3_, CO, NO_2_, PM_10_, and PM_2.5_. Figure 23 shows the data measured on 26 November in Miravalle station. It is worth noting the sensitivity of the system, which was able to detect peaks in the measurement of suspended particles and SO_2_. Further, we observed the SO_2_ and PM’s concentration as a function of height, see Figure 22.

For comparison, the data were contrasted with international air quality measurement systems and with local air quality stations, and the analysis is shown in Table 9.

Figure 24 shows captured images of the drone where the presence of dust can be seen at the monitoring sites (a) Centro station and (b) Miravalle station. 

## 6. Discussion

The change in resistance with the change in gas concentration is not a linear response.

The results were analyzed and compared between the proposed system and international data base. In particular, we estimated the root mean square error, RMSE, as Figure of Merit to compare the performance of the proposed system with international data bases. This comparison is shown in Table 10. We can note that pollution measures are in the expected, i.e., the results showed measured values are very close to the values reported by fixed monitoring systems implemented.

According to the analysis described in Table 10, the measurement system based on low-cost sensors proved to be a very useful alternative to help contrast and complement the data obtained or missing from other methods of measuring air quality pollutants. For example, monitoring pollution emitted into the atmosphere by factories and farms in order to identify and control sources of emission of polluting gases and solid particles. In addition, the proposed system, mounted on a drone, could be used to determine regions of contamination and delimit affected areas in the event of disasters or environmental contingencies. 

The advantages of low cost technologies and simple implementations, such as LoRa, allow the development of practical devices to measure the air quality using a drone. 

The estimated values of suspended particulate matter present the lowest error between the measurement equipment, the presence of gases according to the stations and the system present little variation between locations. The point measurements of suspended particulate matter made by the system usually have better agreement with the international sources. 

We found that the models that describe the behavior of the system resulted in less variation and a better trend in the measurement of SO_2_, CO, NH_2_, in contrast to international monitoring systems. In the case of particulate matter, the proposed system presented unstable measurements with wide fluctuations which were due to the location of the monitoring site. Such is the case of the output values of the PM sensor, which presented some interference in the measurements as a fixed station in Guadalajara and Gomez Palacio. These interferences were punctually focused: cigarette smoke, street sweepers, etc., which at ground level are very punctual. However, the trend and forecast was comparable to the other monitoring systems.

Although that NH_3_ is not measured by any environmental agency we found it is present in Region Lagunera, which is emitted by the agricultural areas and stables, according to [8]. For that, Systems similar to the one proposed in this paper could be used to monitor gases that are not regularly measured by fixed gas monitoring stations. 

In this study, it was important to measure the meteorological parameters in order to calibrate the system and the possibility of correcting the error in the measurements that are directly influenced by them. 

For our future work, we would verify a reference for the NH_3_ measure. Further, in the new design we would be contemplating the use of sensors CO, CO_2_ and O_3_, CO with high sensitivity and capable of detecting low concentration as 122 µg/m^3^. In addition, we would continue to acquire data and verify stability and durability for long-duration operation of the proposed system.

## 7. Conclusions

In this work, we show a remote sensing system designed to monitor the air quality at selected zones, with a particular interest in bigger cities and heavily populated industrial zones, as an alternative to fixed air quality monitoring stations. In particular, we present and discuss the performance of a smart multi-sensor system. The integration of the multi-sensor device, UAV, and LoRa communications as a single, low cost, size and weight system adds a needed flexibility to current fixed monitoring stations with the possibility of mobile monitoring for a larger area and difficult access, since the sensor node can be implemented as end-device in a WSN to obtain data in-situ and real-time.

The multi-sensor system was probed to monitoring sites to determine spatial and temporal patterns of NO_2_, SO_2_, NH_3_, CO, PM_10_, and PM_2.5_ in three important metropolitan areas: Region Lagunera, Monterrey and Guadalajara in Mexico. In addition, the multi-sensor system, acquires measurements of environmental variables such as temperature, humidity, pressure, and wind speed for its calibration and operation. 

The results showed measurement values very close to the values delivered by fixed monitoring systems implemented with complex and expensive technology.

For the future work, we have considered the need to increase measurements to achieve more robust forecast results. In addition, we could implement a WSN to expand the spectrum of environmental monitoring.

The finding of the study can help government agencies, health ministries, and policymakers globally to take proactive actions. The pollution data could be considered by the local air quality regulations in order to prevent risks to the main population in Region Lagunera.

## Figures and Tables

**Figure 1 sensors-22-01706-f001:**
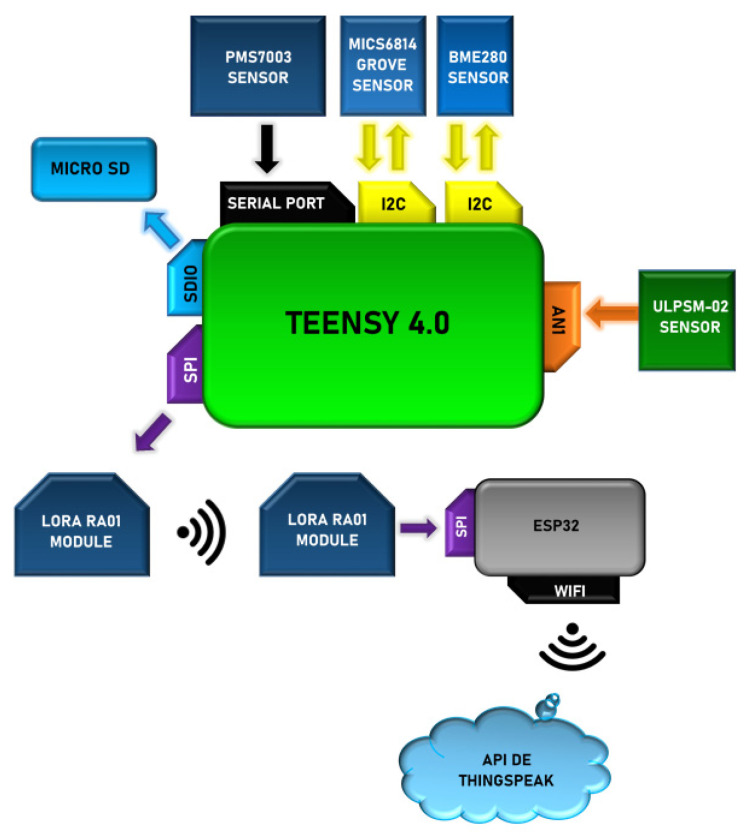
The architecture of smart multi-sensor systems.

**Figure 2 sensors-22-01706-f002:**
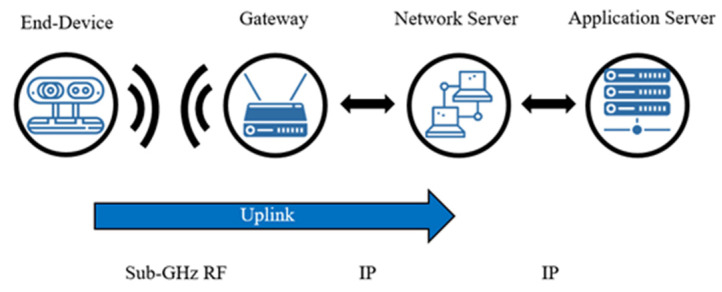
Uplink process.

**Figure 3 sensors-22-01706-f003:**
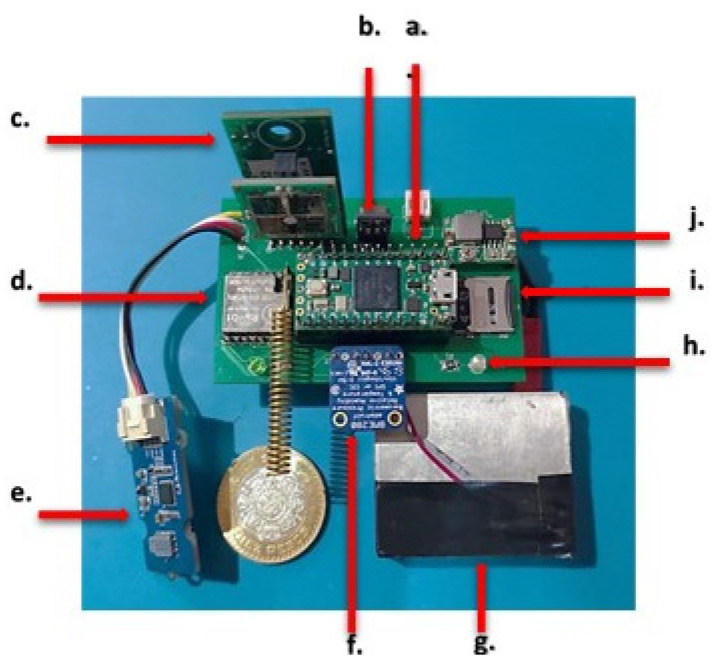
Design of the PCB and devices for the smart multi-sensor node; (**a**) Microcontroller; (**b**) Regulator 3.3 V; (**c**) SO_2_ Gas sensor; (**d**) LoRa WAN; (**e**) Multi-gas sensor; (**f**) Meteorological sensor; (**g**) Particulate sensor; (**h**) Indicator LED’s; (**i**) MicroSD; (**j**) Adjustable voltage regulator.

**Figure 4 sensors-22-01706-f004:**
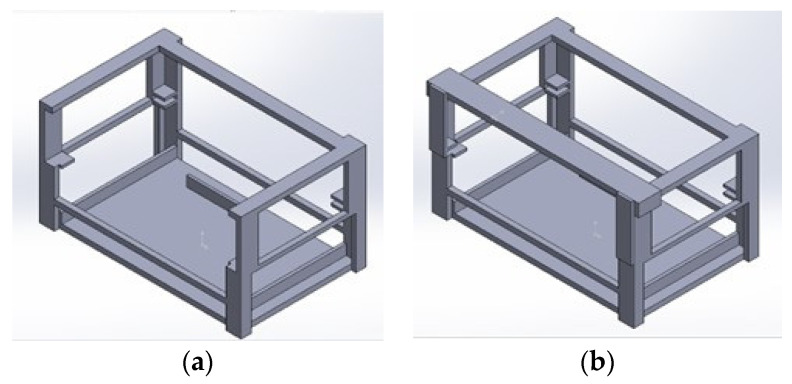
Design of the support box for the multi-sensor system. (**a**) Support base, (**b**) Assembly with support lock.

**Figure 5 sensors-22-01706-f005:**
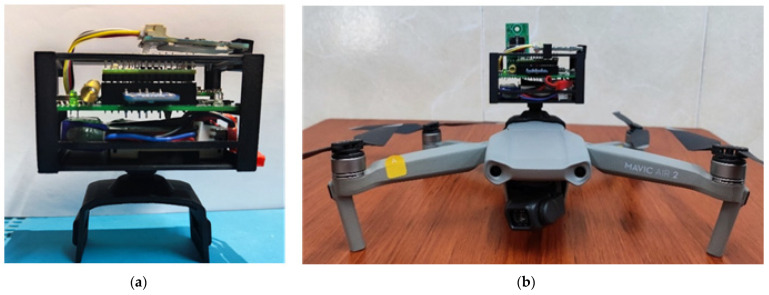
Assembly of the support structure; (**a**) System on the base support; (**b**) Multi-sensor system mounted on the DJI Mavic Air 2 UAV.

**Figure 6 sensors-22-01706-f006:**
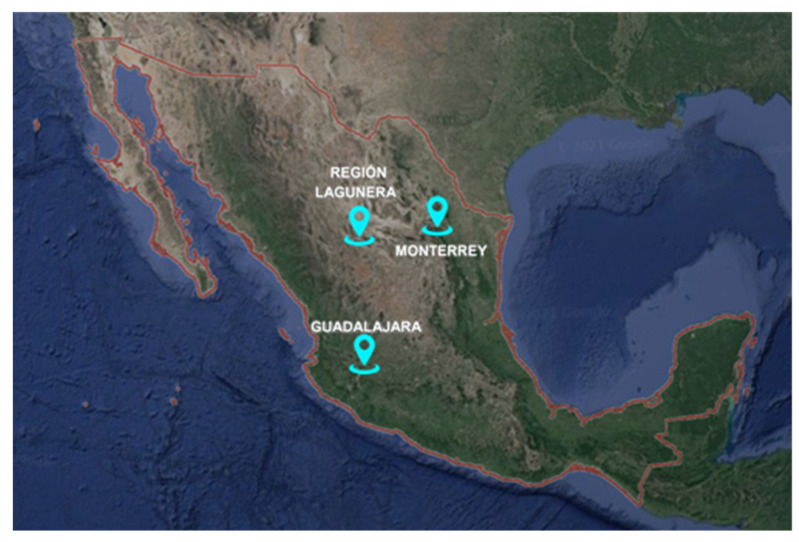
Locations of monitoring sites in Mexico.

**Figure 7 sensors-22-01706-f007:**
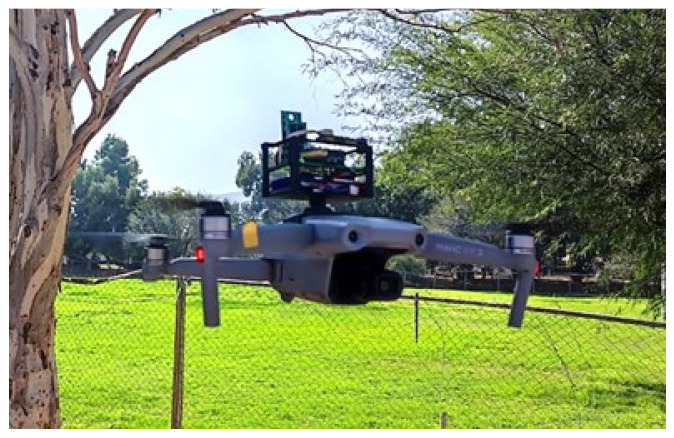
Airborne multi-sensor system flight operation.

**Figure 8 sensors-22-01706-f008:**
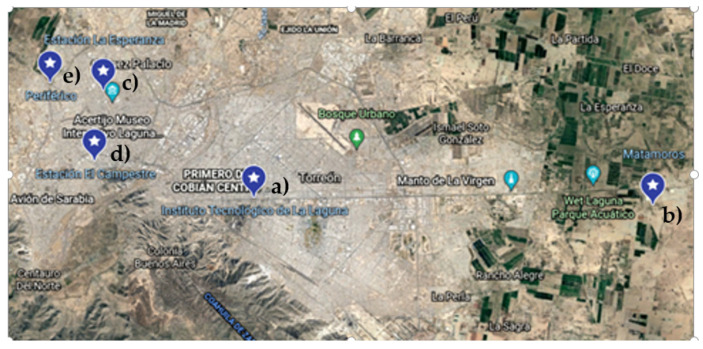
Location of monitoring sites in Region Lagunera: (**a**) Torreon-ITL, (**b**) Matamoros, (**c**) Periferico, (**d**) Campestre monitoring Station, (**e**) La Esperanza Station monitoring.

**Figure 9 sensors-22-01706-f009:**
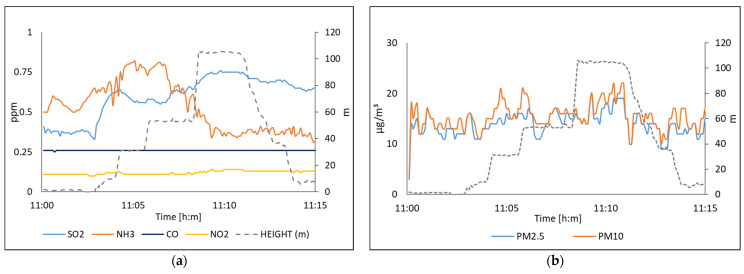
The air quality measured on 11 January in Torreon, Mexico; (**a**) Polluting gases; (**b**) Suspended particles [38].

**Figure 10 sensors-22-01706-f010:**
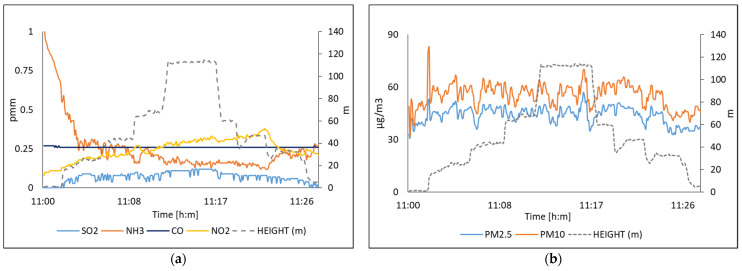
The air quality was measured on 15 January 2021 in Torreon, Mexico; (**a**) Polluting gases; (**b**) Suspended particles [38].

**Figure 11 sensors-22-01706-f011:**
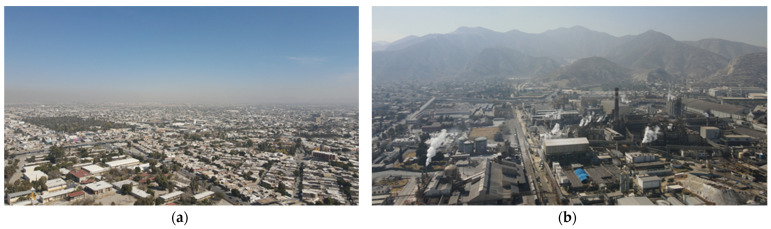
Aerial views of the city of Torreon, Coahuila taken by DJI Mavic Air 2 UAV at an altitude of 100 m; (**a**) Torreon-ITL zone, (**b**) Peñoles.

**Figure 12 sensors-22-01706-f012:**
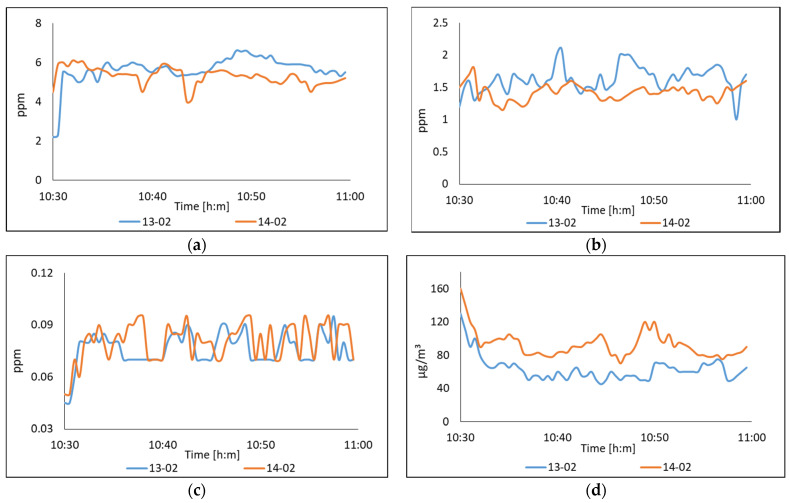
The air quality was measured on crop fields in Matamoros, Coahuila, Mexico. Measured polluting gases were: (**a**) CO; (**b**) NH_3_; (**c**) NO_2_; and (**d**) suspended particles PM_2.5_.

**Figure 13 sensors-22-01706-f013:**
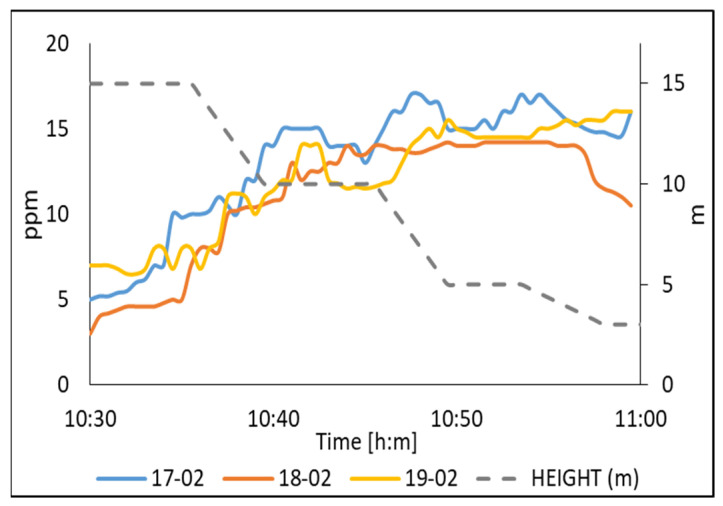
The air quality measured of NH_3_ on a livestock barn in Matamoros, Coahuila, Mexico.

**Figure 14 sensors-22-01706-f014:**
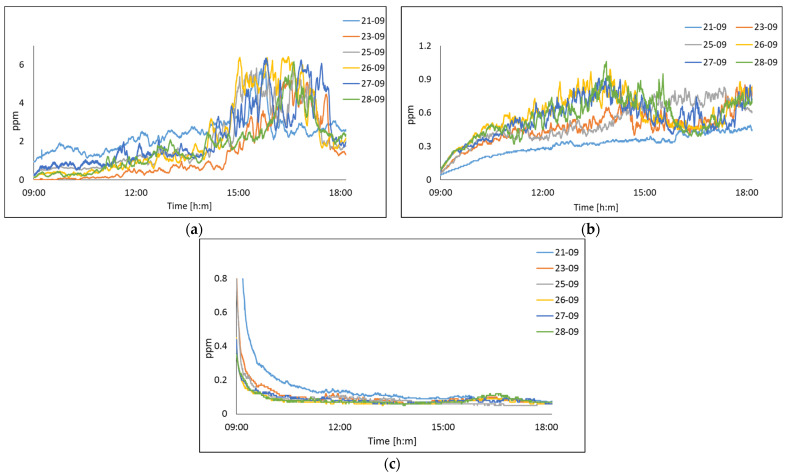
The air quality measured on 21–28 September in Gomez Palacio location of pollutants; (**a**) SO_2_; (**b**) NO_2_; (**c**) NH_3_.

**Figure 15 sensors-22-01706-f015:**
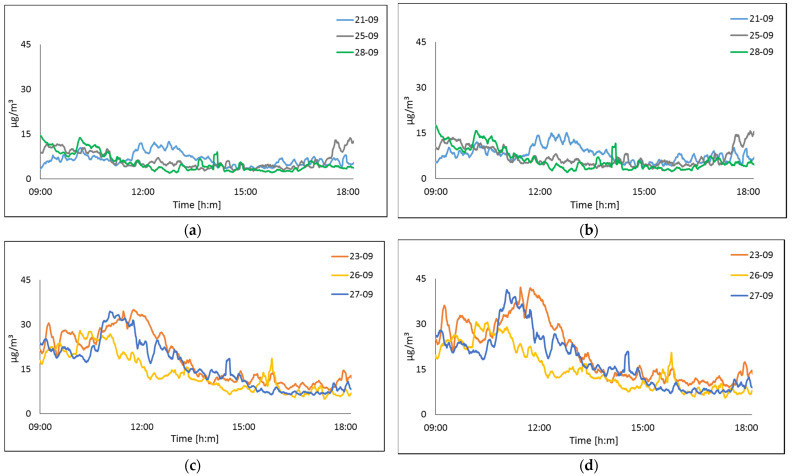
The air quality measured in Gomez Palacio of particulate matter on: (**a**) PM_2.5_ 21, 25 and 28 September; (**b**) PM_10_ 21, 25 and 28 September; (**c**) PM_2.5_ 23, 26 and 27 September; (**d**) PM_10_ 23, 26 and 27 September.

**Figure 16 sensors-22-01706-f016:**
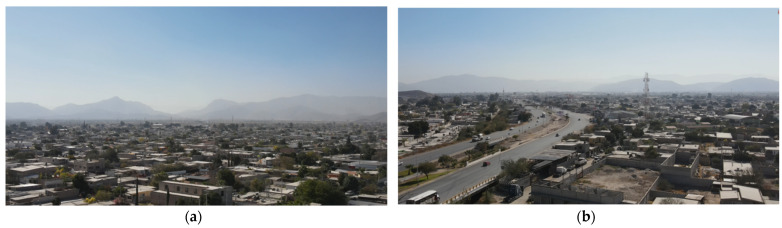
Aerial views of the city of Gomez Palacio, Durango taken by DJI Mavic Air 2 UAV at an altitude of 100 m; (**a**) Periferico zone, 23 September, (**b**) Periferico zone, 27 September.

**Figure 17 sensors-22-01706-f017:**
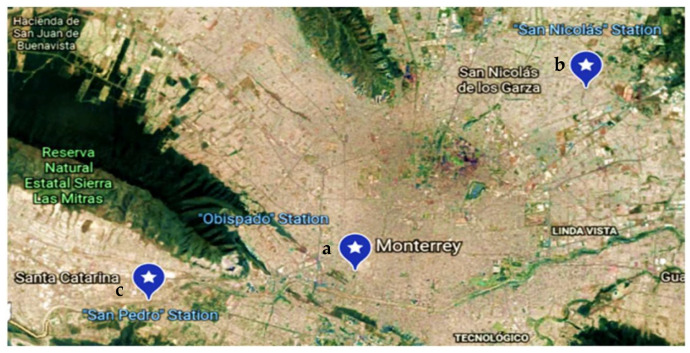
Location of monitoring sites in Monterrey metropolitan area: (**a**) Obispado station monitoring, (**b**) San Nicolas Station monitoring, (**c**) San Pedro station monitoring.

**Figure 18 sensors-22-01706-f018:**
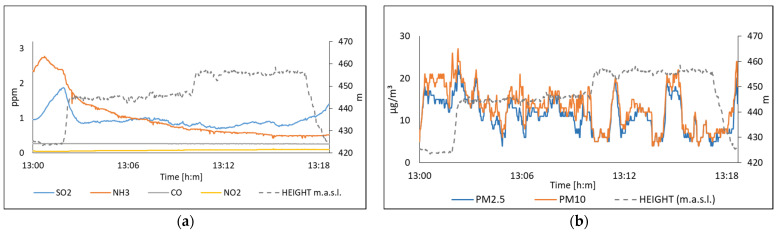
The air quality measured on 15 July, in San Nicolas station; (**a**) Polluting gases; (**b**) Suspended particles.

**Figure 19 sensors-22-01706-f019:**
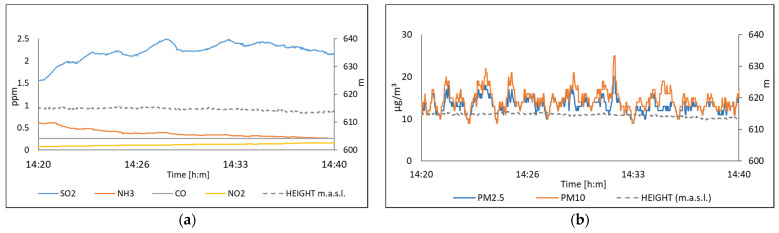
The air quality measured on 16 July, in San Pedro station; (**a**) Polluting gases; (**b**) Suspended particles.

**Figure 20 sensors-22-01706-f020:**
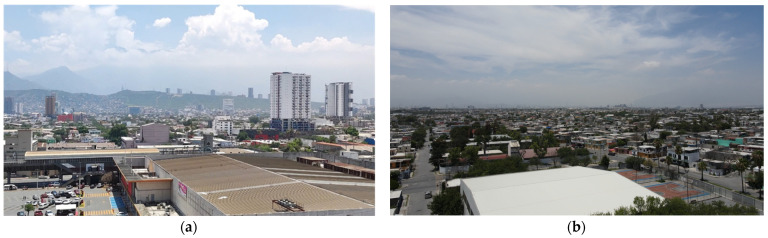
Aerial views of the city of Monterrey, Nuevo Leon taken by DJI Mavic Air 2 UAV at an altitude of 100 m; (**a**) Obispado station, (**b**) San Nicolas station.

**Figure 21 sensors-22-01706-f021:**
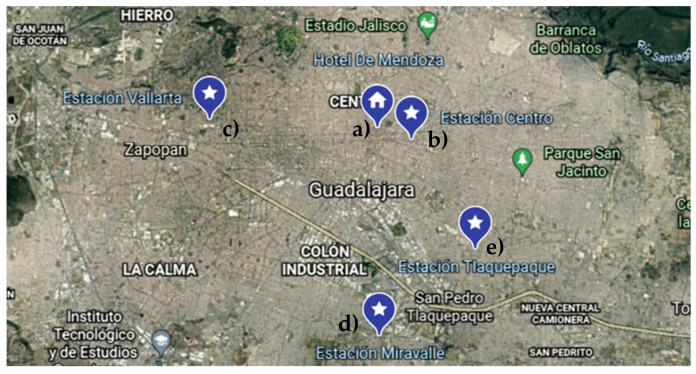
Location of monitoring sites in Guadalajara metropolitan area: (**a**) Mendoza Hotel; (**b**) Centro Station monitoring; (**c**) Vallarta station monitoring; (**d**) Miravalle station monitoring; (**e**) Tlaquepaque station monitoring.

**Figure 22 sensors-22-01706-f022:**
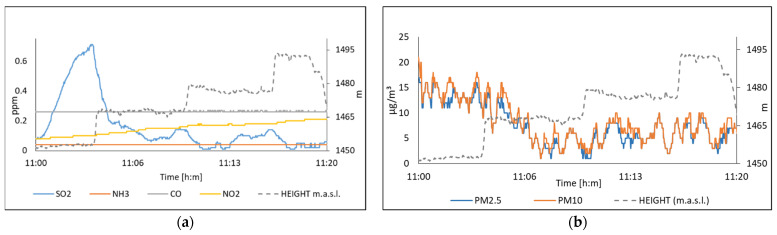
The air quality measured on 24 November, in Centro station; (**a**) Polluting gases; (**b**) Suspended particles.

**Figure 23 sensors-22-01706-f023:**
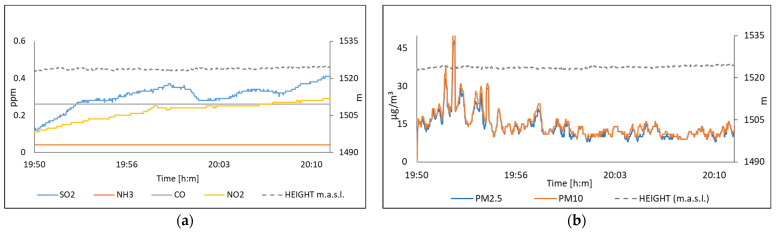
The air quality measured on 26 November, in Tlaquepaque station; (**a**) Polluting gases; (**b**) Suspended particles.

**Figure 24 sensors-22-01706-f024:**
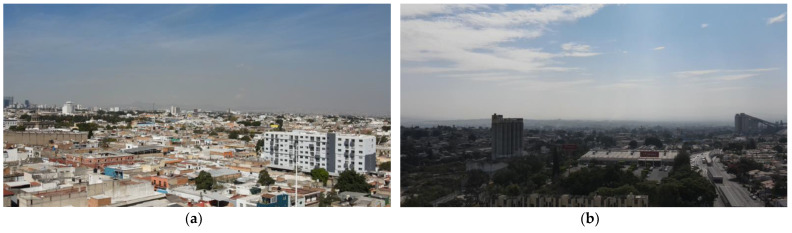
Aerial views of the city of Guadalajara, Jalisco taken by DJI Mavic Air 2 UAV at an altitude of 100 m; (**a**) Centro station, (**b**) Miravalle station.

**Table 1 sensors-22-01706-t001:** Air quality index basics [28].

Index Color	Level of Health Concern	Numeric Value	Meaning
Green	Good	0–50	Air quality is satisfactory, and air pollution poses little or no risk.
Yellow	Moderate	51–100	Air quality is acceptable. However, there may be a risk for some people, particularly those unusually sensitive to air pollution.
Orange	Unhealthy for sensitive groups	100–150	Members of sensitive groups may experience health effects. The general public is less likely to be affected.
Red	Unhealthy	151–200	Some members of the general public may experience health effects; members of sensitive groups may experience more serious health effects.
Purple	Very unhealthy	201–300	Health Alert: The risk of health effects is increased for everyone.
Maroon	Hazardous	301–500	Health Warning of emergency conditions: Everyone is more likely to be affected.

**Table 2 sensors-22-01706-t002:** Information of meteorological parameters [38].

Meteorological Parameter	Date 11 January 2021		Date 15 January 2021	
	Multi-Sensor	Gov. Website	Multi-Sensor	Gov. Website
Temperature (°C)	7	8	16	12
Relative humidity (%)	60	55	22	25
Atmospheric Pressure (hPa)	879	880	878	880
Weather	cold		windy	
Altitude (m.a.s.l.)	1120–1220 ^1^	1120	1120–1220 ^1^	1120
Win speed (m/s)	-	2	-	1–3
Direction speed	-	East	-	West

^1^ Indicates the range of height from home to +100 m provided by drone, where 1120 m.a.s.l. is the altitude in Torreon.

**Table 3 sensors-22-01706-t003:** Comparative values of others monitoring sources and the multi-sensor system [38].

Pollutants	Date: 11 January 2021			Date: 15 January 2021		
	Weather.com	Plumelabs.com	Multi-Sensor	Weather.com	Plumelabs.com	Multi-Sensor
O_3_ (ppm)	0.045	0.044	-	0.051	0.052	-
NO_2_ (ppm)	0.03	0.02	0.12	0.05	0.04	0.25
SO_2_ (ppm)	0.15	-	0.6	0.014	-	0.07
CO (ppm)	0.065	-	0.26	0.052	-	0.26
NH_3_ (ppm)	-	-	0.51	-	-	0.26
PM_10_ (µg/m^3^)	16.5	16	15.55	16.5	16	55
PM_2.5_ (µg/m^3^)	9.72	9	14	10.72	10	43.89

**Table 4 sensors-22-01706-t004:** Information of meteorological parameters in Gomez Palacio, Durango, Mexico.

**Meteorological Parameter**	**Date 21 September 2021**		**Date 23 September 2021**	
	**Multi-Sensor**	**Gov. Website**	**Multi-Sensor**	**Gov. Website**
Temperature (°C)	36	30	29.69	27
Relative humidity (%)	38.78	40	34.4	30
Atmospheric Pressure (hPa)	891.6	890	891.3	890
Weather	Hot		Hot	
Altitude (m.a.s.l.)	1080–1180 ^1^	1135	1080–1180 ^1^	1135
Win speed (m/s)	-	5.28	-	2.2
Direction speed	-	East	-	East
**Meteorological Parameter**	**Date 25 September 2021**		**Date 26 September 2021**	
Temperature (°C)	34	32	34	30
Relative humidity (%)	22	24	39	40
Atmospheric Pressure (hPa)	889	890	890.3	890
Weather	Hot and dry		Hot	
Altitude (m.a.s.l.)	1080–1180 ^1^	1135	1080–1180 ^1^	1135
Win speed (m/s)	-	3.9	-	2.5
Direction speed	-	East	-	West
**Meteorological Parameter**	**Date 27 September 2021**		**Date 28 September 2021**	
Temperature (°C)	35	27	33	30
Relative humidity (%)	28.9	30	31	30
Atmospheric Pressure (hPa)	889.66	890	887.43	890
Weather	Sunny		Sunny	
Altitude (m.a.s.l.)	1080–1180 ^1^	1135	1080–1180 ^1^	1135
Win speed (m/s)	-	1.4	-	2.9
Direction speed	-	West	-	NorthEast

^1^ Indicates the range of height from home to +100 m provided by drone, where 1080 m.a.s.l. is the altitude in Gomez Palacio.

**Table 5 sensors-22-01706-t005:** Comparative values of others monitoring sources and the multi-sensor system.

	**21 September 2021**			**23 September 2021**		
**Pollutants**	**Plume Labs**	**Weather**	**Multi-Sensor**	**Plume Labs**	**Weather**	**Multi-Sensor**
SO_2_ (ppm)	-	0.83	2.41	-	2.89	1.385
NH_3_ (ppm)	-	-	0.214	-	-	0.1
CO (ppm)	-	0.123	0.24	-	0.181	0.25
NO_2_ (ppm)	1.89	1.93	0.317	0.8	0.73	0.495
PM_2.5_ (µg/m^3^)	6.61	7.15	6.5	8.1	8.39	17.98
PM_10_ (µg/m^3^)	13.95	12.95	7.96	20.94	12.22	20.26
	**25 September 2021**			**26 September 2021**		
**Pollutants**	**Plume Labs**	**Weather**	**Multi-Sensor**	**Plume Labs**	**Weather**	**Multi-Sensor**
SO_2_ (ppm)	-	0.81	2.15	-	0.54	2.15
NH_3_ (ppm)	-	-	0.077	-	-	0.077
CO (ppm)	-	0.135	0.26	-	0.233	0.25
NO_2_ (ppm)	0.76	0.76	0.609	0.92	0.54	0.609
PM_2.5_ (µg/m^3^)	6.61	5.85	6.5	10.22	5.16	12.99
PM_10_ (µg/m^3^)	19.88	9.99	7.77	26.16	8.54	14.28
	**27 September 2021**			**28 September 2021**		
**Pollutants**	**Plume Labs**	**Weather**	**Multi-Sensor**	**Plume Labs**	**Weather**	**Multi-Sensor**
SO_2_ (ppm)	-	0.89	2.15	-	1.82	1.77
NH_3_ (ppm)	-	-	0.084	-	-	0.084
CO (ppm)	-	0.133	0.26	-	0.132	0.25
NO_2_ (ppm)	1.03	0.57	0.609	1.44	0.66	0.57
PM_2.5_ (µg/m^3^)	10.22	5.16	12.99	10.77	7	6.33
PM_10_ (µg/m^3^)	26.16	8.54	14.28	18.27	10.67	7.35
	**27 September 2021**			**28 September 2021**		
**Pollutants**	**Plume Labs**	**Weather**	**Multi-Sensor**	**Plume Labs**	**Weather**	**Multi-Sensor**
SO_2_ (ppm)	-	0.89	2.15	-	1.82	1.77
NH_3_ (ppm)	-	-	0.084	-	-	0.084
CO (ppm)	-	0.133	0.26	-	0.132	0.25
NO_2_ (ppm)	1.03	0.57	0.609	1.44	0.66	0.57
PM_2.5_ (µg/m^3^)	10.22	5.16	12.99	10.77	7	6.33
PM_10_ (µg/m^3^)	26.16	8.54	14.28	18.27	10.67	7.35

**Table 6 sensors-22-01706-t006:** Information of meteorological parameters in Monterrey metropolitan area.

Meteorological Parameter	Date 14 July 2021		Date 15 July 2021	
	Multi-Sensor	Gov. Website	Multi-Sensor	Gov. Website
Temperature (°C)	29.47	28.42	30.45	30.45
Relative humidity (%)	48.51	50	45.69	56
Atmospheric Pressure (hPa)	952.4	913	960.8	961
Weather	cloudy		cloudy	
Altitude (m.a.s.l.)	520	540	425–455 ^1^	512
Win speed (m/s)	-	3.167	-	4.63
Direction speed	-	North West	-	West
	**Date 16 July 2021**		**Date 17 July 2021**	
Temperature (°C)	35.48	30.81	31.29	29.92
Relative humidity (%)	39.49	49	42.72	43
Atmospheric Pressure (hPa)	941.68	950	949.07	950
Weather	cloudy		cloudy	
Altitude (m.a.s.l.)	615	540	550	540
Win speed (m/s)	-	5.97	-	2.94
Direction speed	-	North West	-	North

^1^ Indicates the range of height from home to +50/20/10 m provide by drone, where 512 m.a.s.l. is the altitude in Monterrey stations.

**Table 7 sensors-22-01706-t007:** Comparative values of others monitoring sources and the multi-sensor system.

**Pollutants**	**Date: 14 July 2021, Downtown Station**				**Date: 15 July 2021, San Nicolas Station**			
	**Downtown Station (Sinaica)**	**Plumelabs**	**Weather**	**Multi-Sensor**	**San Nicolas Station (Sinaica)**	**Plumelabs**	**Weather**	**Multi-Sensor**
NO_2_ (ppm)	0.003	0.01	0	0.095	0.006	0.01	0	0.08
SO_2_ (ppm)	2	-	2.25	0.9309	4	-	3.9	0.96
CO (ppm)	0.196	-	0.158	0.263	0.204	-	0.106	0.267
NH_3_ (ppm)	-	-	-	1.255	-	-	-	1
PM_10_ (μg/m^3^)	46	15	12.55	9.94	64	6	10.24	13.1
PM_2.5_ (μg/m^3^)	10	9	7.62	8.5	17	4	4.7	11.11
**Pollutants**	**Date: 16 July 2021, San Pedro Station**				**Date: 17 July 2021, Obispado Station**			
	**San Nicolas Station (Sinaica)**	**Plumelabs**	**Weather**	**Multi-Sensor**	**Obispado Station (Sinaica)**	**Plumelabs**	**Weather**	**Multi-Sensor**
NO_2_ (ppm)	0.014	0.01	0	0.14	0.003	0.01	0	0.16
SO_2_ (ppm)	4	-	2.18	2.24	2	-	3.8	1.24
CO (ppm)	0.81	-	0.327	0.26	0.2	-	0.236	0.26
NH_3_ (ppm)	-	-	-	0.32	-	-	-	3
PM_10_ (μg/m^3^)	83	21	22.5	13.88	65	20	15.75	23
PM_2.5_ (μg/m^3^)	19	13	15	12.73	25	12	10.53	20

**Table 8 sensors-22-01706-t008:** Information of meteorological parameters in Guadalajara metropolitan area.

Meteorological Parameter	Date 23 November 2021		Date 24 November 2021		Date 25 November 2021		Date 26 November 2021		Date 27 November 2021	
	Multi-Sensor	Gov. Website	Multi-Sensor	Gov. Website	Multi-Sensor	Gov. Website	Multi-Sensor	Gov. Website	Multi-Sensor	Gov. Website
Temperature (°C)	24	25	27.31	23	26.17	22	23.79	22	26.48	25
Relative humidity (%)	41.12	42	25.62	32	14.12	10	38.95	38	31.49	35
Atmospheric Pressure (hPa)	845.75	1020	849.3	1021	841.88	1020	845	1021	843.05	1020
Weather	Clear sky		Cloudy		cloudy		cloudy		cloudy	
Altitude (m.a.s.l.)	1498	1566	1450–1490 ^1^	1566	1535	1566	1495–1540 ^1^	1566	1524.49	1566
Win speed (m/s)	-	3.68	-	4.33	-	2.52	-	0.48		4.49
Speed direction		SW		S		SE		SE		S

^1^ Indicates the range of height from home to +40 m provide by drone, where 1566 m.a.s.l. is the altitude in Guadalajara, Mex.

**Table 9 sensors-22-01706-t009:** Comparative values of others monitoring sources and the multi-sensor system.

**Pollutants**	**Date: 23 November 2021, Guadalajara, Down Town**				**Date: 24 November 2021, Centro Station**			
	**Downtown Station (Sinaica)**	** plumelabs.com **	** weather.com **	**Sensor System**	**Centro Station (Sinaica)**	** plumelabs.com **	** weather.com **	**Sensor System**
NO_2_ (ppm)	0.015	0	0	0.223	0.009	0	0.96	0.19
SO_2_ (ppm)	0.1	-	0.1	0.105	0.3	-	0.89	0.735
CO (ppm)	0.77	-	0.18	0.26	0.85	-	0.12	0.26
NH_3_ (ppm)	-	-	-	0.04	-	-	-	0.04
PM_10_ (μg/m^3^)	72	19	19.11	20.97	71	10	10.66	6.86
PM_2.5_ (μg/m^3^)	15	27	13	19.49	15	14	7.33	6.14
**Pollutants**	**Date: 25 November 2021, Vallarta Station**				**Date: 26 November 2021, Miravalle Station**			
	**Vallarta Station (Sinaica)**	** plumelabs.com **	** weather.com **	**Sensor System**	**Miravalle Station (Sinaica)**	** plumelabs.com **	** weather.com **	**Sensor System**
NO_2_ (ppm)	-	0	0	0.278	0.14	0.01	0	0.259
SO_2_ (ppm)	0.1	-	0.11	0.324	0.2	-	0.28	-
CO (ppm)	-	-	0.09	0.26	0.77	-	0.28	0.262
NH_3_ (ppm)	-	-	-	0.04	-	-	-	0.04
PM_10_ (μg/m^3^)	-	8	8.11	6.63	-	12	16.35	31
PM_2.5_ (μg/m^3^)	-	5	4.66	5.96	-	8	11.38	27
**Pollutants**		**Date: 26 November 2021, Tlaquepaque Station**						
		**Tlaquepaque Station (Sinaica)**	** plumelabs.com **		** weather.com **	**Sensor System**		
NO_2_ (ppm)		-	0.01		0	0.2499		
SO_2_ (ppm)		0.2	-		0.28	0.322		
CO (ppm)		-	-		0.12	0.258		
NH_3_ (ppm)		-	-		-	0.039		
PM_10_ (μg/m^3^)		36	15		12.21	13		
PM_2.5_ (μg/m^3^)		-	11		8.45	12		

**Table 10 sensors-22-01706-t010:** Measured errors between our air quality system and international data base.

Pollutant	RMSE			
	Multisensor-Weather	Multisensor-Plumelabs	Multisensor-Sinaica	Weather-Plumelab
NO_2_	0.4505	0.5841	0.1376	0.3652
SO_2_	1.3678	-	0.9782	-
CO	0.1128	-	0.2960	-
NH_3_				-
PM_10_	6.1577	8.4557	37.0752	7.1786
PM_2.5_	6.7284	7.3425	11.4225	4.6608

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
