# Peer review of "Smart Multi-Sensor System for Remote Air Quality Monitoring Using Unmanned Aerial Vehicle and LoRaWAN"

_sensors, 2022, doi:10.3390/s22051706_

Round 1

Reviewer 1 Report

This work proposes a smart multi-sensor system to monitor air quality using an unmanned aerial vehicle and LoRa communication system. The designed multi-sensor system measured CO, NO2, NH3, SO2, PM10, and PM2.5. The integration of the multi-sensor device, UAV, and LoRa communications as a single system, adds needed flexibility to currently fixed monitoring stations.

The topic of this paper is interesting and focuses on a highly relevant topic. Using multi-sensor devices (i.e., UAVs and LoRa communications) is vital and complements others studies with similar techniques. The proposed system is adequately described, and the structure is good. The methodology is clear and transparent, and the research procedures and techniques used are standard and reproducible. The results are clear, well presented, and complemented with pictures, tables, and figures to help in results visualization.

However, the authors failed to impress me with the discussion section. This section was poorly written, and the discussion parts do not support the results (L675-681). The link of NO2 with COVID-19 spread is unclear within the context and background of the study. The authors need to reconsider writing the entire paragraph again (L675-681). Last but not least, I recommend language editing before resubmission.

The paper should be suitable for publication after addressing the comments above and applying these minor revisions.

L15: insert “s”; insert “of” to read “… millions of premature …”

L26: insert “tly” to read “… currently”

L62: delete “that is has”

L87: replace “as” with “of” to read “A total of …”

L109: delete “in” to read “…advanced electronics …”

L116-117: Reads better “Several solutions have been proposed to air quality monitoring and measuring environmental parameters in this sense.”

L118: split sentence to read “… WSM. For example …”

L119-120: delete “and is” to read “…and Humidity) connected to PC to store ...

L120: check citation Kasar A. et al. delete “A.” to read “Kasar et al.”

L121-124: Rephrase; insert “the” to read “Kasar A. et al. presented a low low-cost WSN based Air Pollution System using ZigBee protocol and Arduino to monitor CO2 and NO2 [21]. Agustin Candia et al. proposes low-cost systems for managing air quality in cities based on the LoRaWAN network with low-cost sensors to measure PM10 and PM2.5 [22].

L127: delete “a” to read “… and network topology …”

L128: delete “a” to read “… research monitoring …”

L130: delete “A.” to read “Samad et al. …”

L137-138: check the citations and correct them

L137: insert “the” to read “… based on the internet of …”

L138-146: Consider revising the entire sentence.

L159: “...was developed to …”

L170: replace “for measurement of” with “… to measure temperature …”

L171: insert “ing” and “they” to read “… importance of measuring these variables is because they affect …”

L177: “could be measured.”

L178: insert “a” to read “… with a resolution …”

L205: insert “an” to read “…as an end-device …”

L209: insert “of” to read “… to a maximum of 122 m …”

L210: insert “the” to read “… in the horizontal direction …”

L210: add “s” to read “uses”

L211-212: split sentence to read “uplinks. These pass …”

L219: change “this” to “these modules, …”

L224: add “to” to read “… into the smart multi-sensor system”

L230: replace “in addition to” with “and”

L234: delete “can be”

L251: insert “it” to read “… integrated, and it was …”

L260: add “ed” to read “ … and a supported lock with …”

L290: delete “of the” to read “ Each data …”

L291: “… displayed in an …”

L296, 300-301, 303, 306: check citation and format to follow MDPI style. It repeated in subsequent pages below. Kindly amend throughout the article.

Author Response

REVISOR 1

Comments and Suggestions for Authors

This work proposes a smart multi-sensor system to monitor air quality using an unmanned aerial vehicle and LoRa communication system. The designed multi-sensor system measured CO, NO2, NH3, SO2, PM10, and PM2.5. The integration of the multi-sensor device, UAV, and LoRa communications as a single system, adds needed flexibility to currently fixed monitoring stations.

The topic of this paper is interesting and focuses on a highly relevant topic. Using multi-sensor devices (i.e., UAVs and LoRa communications) is vital and complements others studies with similar techniques. The proposed system is adequately described, and the structure is good. The methodology is clear and transparent, and the research procedures and techniques used are standard and reproducible. The results are clear, well presented, and complemented with pictures, tables, and figures to help in results visualization.

Response: We thank this reviewer for the helpful feedback and positive assessment of our work.

However, the authors failed to impress me with the discussion section. This section was poorly written, and the discussion parts do not support the results (L675-681). The link of NO2 with COVID-19 spread is unclear within the context and background of the study. The authors need to reconsider writing the entire paragraph again (L675-681). Last but not least, I recommend language editing before resubmission.

Response: As suggestion by the referee, we rewritten the discussion section and the manuscript was reediting in order to make it more clear and concise. In the revised version, the paragraph has been rewritten in its entirety (L752-798).

L296, 300-301, 303, 306: check citation and format to follow MDPI style. It repeated in subsequent pages below. Kindly amend throughout the article.

Response: Citations have been relocated in the bibliography as web pages.

The paper should be suitable for publication after addressing the comments above and applying these minor revisions.

Response: Thank you for pointing out these typo errors. We fix all typos.

Reviewer 2 Report

  1. The submitted manuscript need revision.
  2. A low quality of figures and they contents. 

Author Response

REVISOR 2

Comments and Suggestions for Authors

  1. The submitted manuscript need revision.

Response: Thanks for your comments. In general, the manuscript has been review and some paragraphs were rewritten.

  1. A low quality of figures and they contents.

Response: As suggestion by the referee, we improved the quality of some figures. While, the data were synthetized seeking to clarify and highlight the results. In particular, in new version of the manuscript we can find:

The content of Figs. 14 to 18 was synthesized for a better visualization of the acquired data, showing a trend over the monitoring time. In new manuscript version, they has been numbering as Fig. 14 and Fig. 15.

For the case of the metropolitan areas of Monterrey, N.L. (Figs. 20 to Fig. 23) and Guadalajara, Jal., (Fig. 25 to Fig. 29), acquired data was synthetized, so only two figures are shown: One of them for the multi-sensor system as a fixed station and another one for the mobile station mounted on the drone. In new manuscript version, the new numbering for figure caption are: Figure 18 and Figure 19 for Monterrey and Figure 22 and Figure 23 for Guadalajara.

Reviewer 3 Report

The work is interesting and can be of value to the research community. The authors have built an IoT system for monitoring the quality of air with respect to 7 different measurements. The measuring device is airborne thanks to a drone. 

The Introduction section is generally well-written but the English needs to be checked by a native speaker. 
Add some sentences on the limitations of the experimental research systems that also use drones in the literature reviews. Comment on how you system improves on these.

Many of the readers will be interested in the hardware for building a similar system. Please include some more information on this. 

There are many graphs of results but virtually no statistical analysis of the signal. What is the variance? How does it compare with the established sensors. How has it changed over time with the reported values. 
Is the measurement of air quality over just 4 days reliable and statistically meaningful?  This should be much longer period. Even a month wont show the seasonal variations that will have an influence on the results. 
How do the results vary with height e.g. 50m or 75m?
What was the weather like (wind direction) on the measurement days.

Many of the URLs are already incorrect. This will get even worse over the time so it is recommended not to use these in the main text of the report. Or put them in the Bibliography. 
The links to some pages are for Spanish audience but it is recommended also to include English ones. 

Add a section commenting on the limitation of the work. 
It would be useful (can be future work) to also have reference measurement in non urban areas. 

Other minor points:

References should be put in format of MDPI format. 
The figures font sizes are too small and should be enlarged. 
No need for titles in the figures or the borders. 

Author Response

REVISOR 3

Comments and Suggestions for Authors

The work is interesting and can be of value to the research community. The authors have built an IoT system for monitoring the quality of air with respect to 7 different measurements. The measuring device is airborne thanks to a drone. 

Response: We thank this reviewer for the helpful feedback and positive assessment of our work.

  • The Introduction section is generally well-written but the English needs to be checked by a native speaker.

Response: The manuscript was extensively revised and corrected according to the comments of the reviewers.

  • Add some sentences on the limitations of the experimental research systems that also use drones in the literature reviews. Comment on how you system improves on these.

Response: Thank you for the helpful comments. In the new version of manuscript, a new paragraph was written, Lines L100-120, where we described how the proposed system improves the limitations of previous developments for monitoring the air quality.

  • Many of the readers will be interested in the hardware for building a similar system. Please include some more information on this.

Response: Thank you for the helpful comments. In new manuscript version, actually he section 3, the description of the proposed system was improved. In particular, we improve the description of the system hardware (L181-186), (L190-196). While, other paragraphs were reorganized in order to improve the clarity in the description of the hardware see (L199-203) and (L213-226).

  • There are many graphs of results but virtually no statistical analysis of the signal. What is the variance? How does it compare with the established sensors. How has it changed over time with the reported values.

Response: The results were analyzed and compared between proposed systems and international data base. In particular, we estimated the root mean square error, RMSE, as Figure of Merit to compare the performance of the proposed system with international data bases. This comparison is shown in table 10. We can note that pollution measures are in the expected, i.e. the results showed measured values are very close to the values reported by fixed monitoring systems implemented. This result was added in discussion.

  • Is the measurement of air quality over just 4 days reliable and statistically meaningful?  This should be much longer period. Even a month won’t show the seasonal variations that will have an influence on the results.

Response: We agree with the review’s comments, for better statistical analyses it is necessary a longer period of monitoring. In particular, considering a fixed station. However, the RMSE depicted in table 10 shows that the performance of the proposed system is similar to that estimated from Weather-Plumelab data bases. Unfortunately, due to the pandemic we were not able to do long-term monitoring. But, we understand the importance of making more prolonged monitoring, in that sense we are planning to install a fixed monitoring station, which takes data for months. This will allow us to make a better correlation between the results obtained by our system and local or international databases. For example, the measurements made for a period of 6 days in Gomez Palacio, Dgo., a trend very similar to that observed in other well established systems. In new manuscript version, a paragraph (L592-600) was written, in which we highlight these observations.

  • How do the results vary with height e.g. 50m or 75m?,

Response: In Fig. 13, Fig. 18 and Fig. 22, one can observe that the values of NH3, SO2, PM10 and PM2.5 show some height dependency. In new manuscript version we write a paragraph (L540-543), (L680-684) and (L729-731), in which we describe briefly this behavior. For example: At lower altitudes, higher concentrations of pollutants, and conversely, at higher altitudes, the concentrations of these pollutants decrease. The paragraph included in the new version of manuscript:

In Figure 13 one observe that NH3 increases as a function of the height when the drone rises. For example, a height of 3m, the NH3 concentration was approximately of 5 ppm. While, at 15 m the NH3 concentration was approximately of 5 ppm. Both measurement are into the permitted level of 25 ppm, above which there are health risks a risk.

In Figure 18, we can be seen that the values of SO2 and NH3 show a decrease as a function of the height, as it is shown in Figure 18a. While in Figure 19 a, the values of SO2 and NH3 show a constant trend at a fixed height. Also, we observed that the SO2 and PM’s concentration as a function of the height, in Figure 22. When the height increases, the concentration of PM's decreases, e.g. from 15ppm to less than 5ppm.

  • What was the weather like (wind direction) on the measurement days.

Response: Yes, we measured the weather conditions see Tables 2, 4, 6 and 8 where the values of: temperature, relative humidity, atmospheric pressure, altitude, direction and speed of the wind, were displayed and they corresponds to the values acquired at the moment of carrying out the measurements by the proposed system. In order to clarify it, in the new manuscript version, a better description was written in paragraph (L576-579) and (L609-611). Also, we described how the change in wind velocity modified the performance of the pollutions.

  • Many of the URLs are already incorrect. This will get even worse over the time so it is recommended not to use these in the main text of the report. Or put them in the Bibliography.

Response: As suggested by the reviewer, the URLs were placed in the bibliography.

The official pages of SINAICA and SMN are not reported in a foreign language. However, state government information pages have been included in the English language.

  • Add a section commenting on the limitation of the work. It would be useful (can be future work) to also have reference measurement in non urban areas.

Response: Future work is described in the discussion Section (L778-786 and  L794-798) and in the conclusions Section (L815-817).

Minor revisions:

References should be put in format of MDPI format. 

Response: Thanks for your suggestion. All the references has been put in MDPI format.
The figures font sizes are too small and should be enlarged. 

Response: Thanks for your suggestion. All the figures were corrected in resolution format.

No need for titles in the figures or the borders.

Response: Thanks for your suggestion. All the titles into the figures were eliminated.

Round 2

Reviewer 2 Report

Paper needs to be proofread to be rid of grammatical errors/omissions.

Reviewer 3 Report

The authors have answered my questions to a satisfactory level. I am satisfied with the updated version of the manuscript There are some minor formatting comments. Authors should check the format for references in the format of MDPI ie Name.Initials.